# Chronic Hepatitis B Virus Infection and HLA Variations in a Greek Population

**DOI:** 10.3390/v17040462

**Published:** 2025-03-24

**Authors:** Evangelia Myserli, Ioannis Goulis, Asimina Fylaktou, Maria Exindari, Fani Minti, Georgia Chatzika, Eleni Iliopoulou, Polyxeni Agorastou, Ioanna Papagiouvanni, Theodora Oikonomou, Georgia Gioula

**Affiliations:** 1Microbiology Department, Medical School, Aristotle University of Thessaloniki, 54124 Thessaloniki, Greece; mexidari@auth.gr (M.E.); fani.minti@yahoo.com (F.M.); ggioula@auth.gr (G.G.); 2Fourth Department of Internal Medicine, Hippokration General Hospital, 54642 Thessaloniki, Greece; igoulis@gmail.com (I.G.); pagorast@gmail.com (P.A.); ioanna.d.pap@gmail.com (I.P.); oikotheod@yahoo.gr (T.O.); 3National Peripheral Histocompatibility Center, Immunology Department, Hippokration General Hospital, 54642 Thessaloniki, Greece; fylaktoumina@gmail.com (A.F.); g.chatzika@hotmail.com (G.C.); iliopoulou.elen@gmail.com (E.I.)

**Keywords:** HLA class I alleles, HLA class II alleles, chronic HBV infection, HBV clearance, HBV virus

## Abstract

Chronic hepatitis B is linked with considerable liver-related morbidity and mortality globally. Human leukocyte antigen (HLA) polymorphisms affect the susceptibility and outcome of many immune-mediated diseases and infections. Our aim was to study the impact of HLA alleles on HVB-infected individuals in a Greek population. In total, 107 patients with chronic HBV infection (cHBV group) and 101 with spontaneous clearance (SC-group) of hepatitis B surface antigen (HBsAg) were genotyped for HLA-A, HLA-B, HLA-C, HLA-DRB1, and HLA-DQB1 loci by single-specific primer polymerase chain reaction (PCR-SSP). The HLA alleles’ frequencies were compared between the two patient groups and healthy individuals from the North Greece Bone Marrow Donor Registry (14506 samples). We found a significantly increased frequency of HLA-C*01 and HLA-DRB1*16 alleles in the cHBV group versus the SC-group. The frequency of HLA-A*01, HLA-B*08, HLA-C*01, HLA-C*08, HLA-DRB1*03, and HLA-DQB1*05 alleles was significantly higher in cHBV patients versus healthy individuals, while the frequency of the HLA-B*38 allele was significantly lower. Our study showed an association of specific HLA alleles with either susceptibility or protection against chronic HBV infection.

## 1. Introduction

Chronic hepatitis B (CHB) is a serious public health problem, affecting 296 million people worldwide [1]. Although hepatitis B usually causes an acute and self-limiting disease, 5% of infected adults and up to 95% of individuals infected in early childhood will acquire chronic HBV infection. HBV persistence can lead to liver cirrhosis and even hepatocellular carcinoma (HCC) [2].

Chronic hepatitis B prevalence varies significantly in different geographic regions, with Central/East Asia, Sub-Saharan Africa, and the Pacific region reporting a prevalence of about 5–8%, in contrast to the United States, where the respective number is close to 0.3% [3]. In Greece, CHB prevalence in the general population is 1.88% but there are cluster populations that present higher rates [4,5]. Host genetic, viral, socioeconomic, and environmental factors contribute to these regional differences in HBV infection incidence and outcomes [6].

The Major Histocompatibility Complex (MHC) encodes the human leukocyte antigens (HLAs) vital in the immune system’s response to different pathogens. HLA class I molecules present endogenous peptides to CD8+ T cells that cytotoxically attack infected cells. In contrast, HLA class II molecules present exogenous antigens, leading to the activation of T helper lymphocytes [7]. The polymorphism of HLA genes at the genome, allele, or haplotype level has been intensively investigated in correlation with various diseases in different populations. According to the literature, several HLA polymorphisms have been associated with either susceptibility or protection against HBV virus persistence. Research in this field would provide more insights into disease etiology, risk assessment, and prognosis. The aim of this study was to assess the prevalence of HLA alleles in patients with chronic HBV infection and compare these findings to those of healthy individuals to determine the association between HLA polymorphism and chronic HBV risk [8,9,10]

## 2. Materials and Methods

We enrolled patients ≥ 18 years of age diagnosed with HBV infection at the outpatient clinic of Hippokration General Hospital of Thessaloniki. This study was approved by the Institutional Review Board of Hippokration General Hospital (decision No. 29075/20-6-2024). All patients provided written informed consent following the Helsinki Declaration and were subsequently enrolled in the study.

The study population was divided into two groups. (a) The spontaneous clearance (SC) group included 101 HBV-infected individuals who spontaneously cleared the hepatitis B surface antigen (HBsAg) and acquired natural immunity to the virus. Laboratory workup in the SC group was consistent with negative HBsAg testing, positive testing for the anti-hepatitis B surface antibody (anti-HBs) and anti-hepatitis B core antibody (anti-HBc), and normal liver function biochemical tests values. (b) The chronic HBV (cHBV) group included 107 individuals with chronic HBV infection. Laboratory workup in the cHBV group consistent with positive HBsAg for at least 6 months before study entry, positive testing for anti-HBc, and undetectable anti-hepatitis B core immunoglobulin M (anti-HBc IgM) antibody. Finally, we used adults with a healthy medical record from the Northern Greece Bone Marrow Donor Registry (14,506 samples) as a control group. Bone marrow donors were between 18 and 50 years old and they did not suffer from asthma, insulin-dependent diabetes, cancer, heart disease, multiple sclerosis, muscular dystrophy, schizophrenia, depression, infectious diseases, such as human immunodeficiency virus (HIV), hepatitis B or C, autoimmune disorders, vascular diseases, arterial or venous thrombosis, or von Willebrand disease. In general, donors had a healthy medical record according to national guidelines for blood marrow donors. Individuals with a history of drug abuse, as well as those who were recipients of solid organ or hematopoietic cells in the past or were at risk for Creutzfeldt–Jacob disease, were excluded from the study.

Exclusion criteria for this study were infection with other hepatotropic viruses, diagnosis of alcoholic hepatitis, steatohepatitis, autoimmune hepatitis, toxic hepatitis, primary biliary cirrhosis, and non-HBV-related HCC.

We collected peripheral blood samples for DNA extraction using the iPrep™ PureLink™ gDNA Blood kit (ThermoFischer Scientific, Waltham, MA, USA). HLA-A, -B, -C, -DRB1, and -DQB1 alleles of the study participants were genotyped in low-resolution analyses by single-specific primer polymerase chain reaction (SSP-PCR). Statistical analysis was performed using R programming language, version 4.2.2. [11] The detected HLA alleles frequencies were compared among the groups of our study with the 2-sided Fisher’s exact test. The Haldane–Anscombe correction was used when needed. Results with *p*-value < 0.05 were considered statistically significant. Odds ratios (OR) with 95% confidence intervals (CI) were calculated to further strengthen the results.

## 3. Results

Patient demographic characteristics are shown in Table 1. In Table 2, we present the HLA analysis using Fisher’s exact test assessing the prevalence of HLA alleles between the SC and cHBV group. Results revealed that the prevalence of alleles HLA-C*01 (8.82% vs. 0.91%, *p* = 0.005 < 0.05, OR = 0.1, 95% CI [0.0—0.62]) and HLA-DRB1*16 (13.73% vs. 7.14%, *p* = 0.03 < 0.05, OR = 0.48, 95% CI [0.23—0.99]) was significantly higher in the cHBV group compared to SC group. No other significant differences regarding the prevalence of HLA alleles were observed between the two groups.

In Table 3, we present the HLA analysis using Fisher’s exact test assessing the prevalence of HLA alleles between the cHBV and control group. Results revealed that the prevalence of alleles HLA-A*01 (16.18% vs. 9.7%, *p* = 5.21 × 10^−10^ < 0.05, OR = 4.33, 95% CI [2.77—6.64]), HLA-B*08 (8.33% vs. 4.3%, *p* = 0.009 < 0.05, OR = 2.02, 95% CI [1.15—3.34]), HLA-C*01(8.82% vs. 4.2%, *p* = 0.01 < 0.05, OR = 2.18, 95% CI [1.15—4.0]), HLA C*08 (5.88% vs. 2.2%, *p* = 0.009 < 0.05, OR = 2.8, 95% CI [1.22—6.12]), HLA-DRB1*03 (10.29% vs. 5.8%, *p* = 0.01 < 0.05, OR = 1.86, 95% CI [1.12—2.95]), and HLA-DQB1*05 (30.69% vs. 22%, *p* = 0.04 < 0.05, OR = 1.57, 95% CI [1.01—2.46]) was significantly higher in the cHBV group compared to healthy individuals group. On the contrary, allele HLA-B*38 (3.4% vs. 0.5%, *p* = 0.017 < 0.05, OR = 0.14, 95% CI [0.0—0.79]), was found in a significantly lower frequency in chronically infected patients versus the healthy individuals group. No other significant differences regarding the prevalence of HLA alleles were found between the two groups.

## 4. Discussion

In this study, we compared the prevalence of HLA alleles between patients with chronic HBV infection and natural immunity to HBV, and we further compared the distribution of HLA alleles between healthy adult individuals and patients with chronic HBV. Both comparisons give complementary insight into the impact of HLA alleles’ polymorphism on HBV infection prognosis and risk, respectively.

Our research showed an association of five (5) specific HLA class I and three (3) HLA class II alleles with HBV infection in a Greek population.

### 4.1. HLA Class I Alleles Associations

Regarding HLA class I associations our results showed that the prevalence of the HLA-C*01 allele was significantly higher in patients with cHBV compared with patients with natural immunity to HBV. Also, compared to healthy adult individuals, patients with cHBV had a significantly higher prevalence of HLA-A*01, HLA-B*08, HLA-C*01, and HLA C*08, and a significantly lower prevalence of HLA-B*38. HLA-C*01 is common between the two comparisons, implying both a predictive and a prognostic role.

Other researchers have also demonstrated the association of HLA class I alleles with susceptibility to or protection from chronic hepatitis B. Thio et al. found that HLA-A*03:01 allele frequency was higher in Caucasian people, while HLA-B*08, both alone and as part of the conserved haplotype A*01-B*08, and HLA-B*44 allele frequencies were lower compared to patients with persistent HBV infection [12]. In their research, Ramezani [12] et al. correlated HLA-A*33 with chronic HBV infection [13] A meta-analysis that included 1652 healthy individuals and 659 chronic HBV patients from different geographic regions showed a protective role of HLA-B*07 and HLA-B*58 against HBV persistence [14]. HLA-C*08:04 have been recently linked with HBV clearance in a Ghanaian cohort [15]. In previous research of our team, HLA-A*01 and B*57 alleles were significantly associated with chronic HBV complications such as cirrhosis and hepatocellular carcinoma (HBV-HCC) while the HLA-C*15 allele was linked with protection against HBV persistence complications [16]. To our knowledge, this is the first time that HLA-C*01 and HLA-B*38 have been associated with HBV infection. Regarding associations with other infections, HLA-C*01 has been linked with susceptibility to COVID-19 in an Italian cohort [17]. HLA-B*38 has been linked either with protection against leishmaniasis [18] or risk of severe COVID-19 infection [19].

### 4.2. HLA Class II Alleles Associations

Regarding HLA class II associations, our results showed that the prevalence of the HLA-DRB1*16 allele was significantly higher in patients with cHBV compared with patients with natural immunity to HBV. Additionally, HLA-DRB1*03 and HLA-DQB1*05 alleles were found in a significantly higher frequency in cHBV patients versus the healthy adults group.

HLA-DRB1*16 was associated with susceptibility to CHB in a Chinese population by Han et al. [20]. Höhler et al.’s study on Caucasian patients demonstrated an association of HLA-DRB1*13:01—02 with the protection of chronic HBV infection [21]. Jin et al. showed an association between the HLA-DRB1*140101 allele and an increased risk of complicated CHB [22]. HLA DRB1*04:03 and HLA-DRB1*15:01 have been linked with antibodies to hepatitis B surface antigen (anti-HBsAg) status [23].

Other studies have shown an association of polymorphisms in the HLA-DQB1*05 locus, such as HLA-DQB1*05:02 [24] and HLA-DQB1*05:03 [25] with the risk of chronic HBV infection. Further polymorphisms in other HLA-DQB1 regions, including HLA-DQB1*02, HL-DQB1*03, and HLA-DQB1*06, are also associated with the persistence of HBV virus [6]. Huang et al.’s meta-analysis, including 815 chronically HBV-infected patients and 731 healthy individuals, associated the HLA-DQB1*02:01, *03:01, and *05:02 alleles with CHB risk [24]. It is also interesting that Matei et al. associated the HLA-DQB1*05 allele and HLA-DRB1*03 with susceptibility to HBV in a European–Caucasian population, as we did [26]. Furthermore, we have found in previous research the presence of the HLA-DQB1*05:01 allele in significantly higher frequency in chronically HBV-infected patients with complications [16].

On the other hand, Tălăngescu et al. [27] found polymorphisms within the HLA-DQB1*01, HLA- DQB1*06, HLA-DQB1*13, and HLA-DQB1*15 regions with a protective role against chronic HBV infection. Also, Naderi et al. [28] revealed a protective role of HLA-DQB1*06:04 polymorphism against viral persistence. In Huang J. et al.’s meta-analysis, HLA-DQB1*0303 and *0604 alleles were also associated with protection against hepatitis B virus persistence [24]. Furthermore, HLA-DQB1 alleles have also been associated with response to HBV vaccination [29].

Further low and high-resolution analysis is critical to accurately define the genetic markers that positively or negatively influence susceptibility to HBV and the clinical outcome of the infected individuals.

Our study’s limitations include the relatively small sample size, due to the eligibility criteria for the chronic HBV patients and the fact that Greece is a low HBV-endemicity country (CHB prevalence 1.88%) [4]. A further limitation is that multivariable analysis for potential inter-group confounders was not available because we were unable to extract the demographic data of healthy subjects, apart from median age and underlying comorbidities that could exclude them from blood marrow donating procedure. Finally, HLA analysis was carried out only in low resolution (2-digit analysis) due to limited funding and shortage of stored DNA samples for further analysis.

## 5. Conclusions

In conclusion, our findings linked HLA-C*01 and HLA-DRB1*16 alleles with worse disease prognosis, since they were found in a statistically significant lower frequency in the HBsAg spontaneous clearance group vs. chronically infected patients. HBsAg spontaneous clearance can lead to biochemical, virological, and liver histological improvement. As a result, these potential biomarkers should be further assessed in relation to HBV-related complications such as HCC [30,31,32,33] We also highlighted HLA-A*01, HLA-B*08, HLA-B*38, HLA-C*01, HLA C*08, HLA-DRB1*03, and HLA-DQB1*05 alleles as potential markers for early detection or preventive measures for at-risk individuals since we found them in a statistically higher frequency in chronically infected patients vs. healthy people. Our findings for alleles HLA-A*01 and HLA-DQA*05 are consistent with the previous results of our team, which show correlation with HBV infection complications such as HBV-cirrhosis and HBV-HCC [16]. HLA allele or SNP frequencies, as well as linkage disequilibrium across the HLA region, vary greatly between populations from different geographic regions, and conflicting data between research in populations of different origins are often reported. To our knowledge, this is the first time that the association of HLA polymorphisms with chronic HBV has been studied in a Greek population.

The identification of factors that could lead to HBV virus persistence is of major importance since they can play a prognostic role or even contribute to early diagnosis and targeted treatment of patients at risk for chronic HBV infection. Validation of the association of such host genetic polymorphisms with HBV infection chronicity should be carried out in further large-scale, and multi-center studies.

## Figures and Tables

**Table 1 viruses-17-00462-t001:** Baseline characteristics of patient groups included in the study.

Variables	cHBV Group (*n* = 107)	SC Group (*n* = 101)
Male, number (%)	86 (80.4)	68 (67.3)
Female, number (%)	21 (19.62)	33 (32.7)
Age (years), mean ± SD	60.6 ± 5.1	60.9 ± 12.0

**Table 2 viruses-17-00462-t002:** HLA analysis between patients in the cHBV and SC group.

Allele	Patients in the cHBV Group (2N = 214)	Patients in the cHBV Group, %	Patients in the SC Group (2N = 202)	Patients in the SCgroup, %	*p* Value	OR	95%CI
**HLA–A**							
A*01	33	16.18	22	11.11	0.15	0.65	[0.35—1.2]
A*02	45	22.06	56	28.28	0.17	1.39	[0.86—2.25]
A*03	22	10.78	13	6.57	0.16	0.58	[0.26—1.25]
A*11	10	4.9	17	8.59	0.16	1.82	[0.76—4.57]
A*23	7	3.43	4	2.02	0.54	0.58	[0.12—2.33]
A*24	30	14.71	31	15.66	0.89	1.08	[0.6—1.93]
A*25	1	0.49	2	1.01	0.62	2.07	[0.11—122.71]
A*26	12	5.88	9	4.55	0.66	0.76	[0.28—2.02]
A*29	6	2.94	3	1.52	0.5	0.51	[0.08—2.42]
A*30	2	0.98	3	1.52	0.68	1.55	[0.18—18.76]
A*31	3	1.47	6	3.03	0.33	2.09	[0.44—13.1]
A*32	13	6.37	13	6.57	1	1.03	[0.43—2.49]
A*33	5	2.45	3	1.52	0.72	0.61	[0.09—3.2]
A*66	1	0.49	3	1.52	0.37	3.12	[0.25—164.61]
A*68	11	5.39	11	5.56	1	1.03	[0.4—2.7]
**HLA–B**							
B*07	6	2.94	4	2.02	0.75	0.68	[0.14—2.94]
B*08	17	8.33	7	3.54	0.06	0.41	[0.14—1.06]
B*13	4	1.96	4	2.02	1	1.04	[0.19—5.65]
B*14	10	4.9	7	3.54	0.62	0.72	[0.23—2.13]
B*15	8	3.92	4	2.02	0.38	0.51	[0.11—1.94]
B*18	17	8.33	24	12.12	0.25	1.52	[0.76—3.13]
B*27	6	2.94	5	2.53	1	0.86	[0.2—3.44]
B*35	37	18.14	31	15.66	0.59	0.84	[0.48—1.47]
B*37	5	2.45	2	1.01	0.45	0.41	[0.04—2.53]
B*38	1	0.49	5	2.53	0.12	5.27	[0.58—250.94]
B*39	5	2.45	5	2.53	1	1.04	[0.23—4.58]
B*40	9	4.41	14	7.07	0.29	1.66	[0.65—4.45]
B*41	4	1.96	4	2.02	1	1.04	[0.19—5.65]
B*44	14	6.86	15	7.58	0.85	1.12	[0.49—2.58]
B*49	3	1.47	3	1.52	1	1.04	[0.14—7.83]
B*50	2	0.98	5	2.53	0.28	2.62	[0.42—27.87]
B*51	28	13.73	30	15.15	0.67	1.13	[0.62—2.05]
B*52	7	3.43	10	5.05	0.46	1.5	[0.5—4.76]
B*55	5	2.45	4	2.02	1	0.83	[0.16—3.9]
B*56	2	0.98	1	0.51	1	0.52	[0.01—9.99]
B*57	6	2.94	4	2.02	0.75	0.68	[0.14—2.94]
B*58	3	1.47	1	0.51	0.62	0.34	[0.01—4.31]
B*73	1	0.49	1	0.51	1	1.04	[0.01—81.65]
**HLA–C**							
**C*01**	**18**	**8.82**	**1**	**0.91**	**0.005**	**0.1**	**[0—0.62]**
C*02	14	6.86	6	5.45	0.81	0.78	[0.24—2.25]
C*03	14	6.86	8	7.27	1	1.06	[0.37—2.83]
C*04	36	17.65	17	15.45	0.75	0.85	[0.42—1.66]
C*05	7	3.43	4	3.64	1	1.06	[0.22—4.29]
C*06	16	7.84	9	8.18	1	1.05	[0.39—2.62]
C*07	37	18.14	18	16.36	0.76	0.88	[0.45—1.7]
C*08	12	5.88	5	4.55	0.8	0.76	[0.2—2.4]
C*12	25	12.25	17	15.45	0.49	1.31	[0.63—2.67]
C*14	2	0.98	2	1.82	0.61	1.87	[0.13—26.08]
C*15	12	5.88	14	12.73	0.05	2.33	[0.96—5.74]
C*16	6	2.94	6	5.45	0.36	1.9	[0.49—7.3]
C*17	4	1.96	2	1.82	1	0.93	[0.08—6.58]
**HLA–DRB1**							
DRB1*01	21	10.29	23	11.73	0.75	1.16	[0.59—2.29]
DRB1*03	21	10.29	17	8.67	0.61	0.83	[0.4—1.71]
DRB1*04	12	5.88	19	9.69	0.19	1.72	[0.77—3.99]
DRB1*07	13	6.37	9	4.59	0.51	0.71	[0.26—1.84]
DRB1*10	2	0.98	4	2.04	0.44	2.1	[0.3—23.47]
DRB1*11	61	29.9	55	28.06	0.74	0.91	[0.58—1.44]
DRB1*12	2	0.98	7	3.57	0.1	3.73	[0.7—37.27]
DRB1*13	15	7.35	22	11.22	0.23	1.59	[0.76—3.41]
DRB1*14	14	6.86	12	6.12	0.84	0.89	[0.36—2.12]
DRB1*15	13	6.37	13	6.63	1	1.04	[0.43—2.52]
**DRB1*16**	**28**	**13.73**	**14**	**7.14**	**0.03**	**0.48**	**[0.23—0.99]**
**HLA–DQB1**							
DQB1*02	31	15.35	7	11.67	0.54	0.73	[0.26—1.82]
DQB1*03	82	40.59	33	55	0.05	1.78	[0.96—3.34]
DQB1*05	62	30.69	16	26.67	0.63	0.82	[0.4—1.62]
DQB1*06	25	12.38	4	6.67	0.25	0.51	[0.12—1.56]

Note: statistically significant associations are presented in bold. Abbreviations: cHBV—chronic hepatitis B virus; CI—confidence interval; HLA—human leucocyte antigen; SC—spontaneous clearance.

**Table 3 viruses-17-00462-t003:** HLA analysis between cHBV and healthy adult group.

Allele	Patients in the cHBV Group (2N = 214)	Patients in the cHBV Group, %	Healthy Adults (2N = 29,012)	Healthy Adults, %	*p*-Value	OR	95%CI
**HLA–A**							
**A*01**	**33**	**16.18**	**2814**	**9.7**	**5.21** **× 10****^−^****^10^**	**4.33**	**[2.77—6.64]**
**A*02**	45	22.06	7921	27.3	0.1	0.75	[0.53—1.06]
**A*03**	22	10.78	2698	9.3	0.47	1.18	[0.72—1.84]
**A*11**	10	4.9	2002	6.9	0.33	0.7	[0.33—1.31]
**A*23**	7	3.43	928	3.2	0.84	1.08	[0.43—2.27]
**A*24**	30	14.71	4497	15.5	0.85	0.94	[0.61—1.39]
**A*26**	12	5.88	1625	5.6	0.88	1.05	[0.53—1.89]
**A*32**	13	6.37	1770	6.1	0.88	1.05	[0.55—1.84]
**A*68**	11	5.39	1161	4	0.28	1.37	[0.67—2.51]
**HLA–B**							
**B*07**	6	2.94	1230	4.2	0.48	0.69	[0.25—1.54]
**B*08**	**17**	**8.33**	**1259**	**4.3**	**0.009**	**2.02**	**[1.15—3.34]**
**B*18**	17	8.33	3396	11.6	0.19	0.69	[0.39—1.14]
**B*35**	37	18.14	5446	18.6	0.93	0.97	[0.66—1.39]
**B*38**	**1**	**0.49**	**995**	**3.4**	**0.017**	**0.14**	**[0—0.79]**
**B*40**	9	4.41	966	3.3	0.33	1.35	[0.61—2.63]
**B*44**	14	6.86	2196	7.5	0.89	0.91	[0.49—1.57]
**B*51**	28	13.73	4157	14.2	0.92	0.96	[0.62—1.44]
**HLA–C**							
**C*01**	**18**	**8.82**	**38**	**4.2**	**0.01**	**2.18**	**[1.15—4.0]**
**C*02**	14	6.86	58	6.3	0.75	1.09	[0.55—2.03]
**C*03**	14	6.86	46	5.0	0.3	1.39	[0.69—2.64]
**C*04**	36	17.65	156	17.0	0.83	1.04	[0.68—1.57]
**C*05**	7	3.43	20	2.2	0.31	1.59	[0.56—3.98]
**C*06**	16	7.84	88	9.6	0.5	0.8	[0.43—1.41]
**C*07**	37	18.14	164	17.9	0.92	1.02	[0.67—1.52]
**C*08**	**12**	**5.88**	**20**	**2.2**	**0.009**	**2.8**	**[1.22—6.12]**
**C*12**	25	12.25	164	17.9	0.06	0.64	[0.39—1.02]
**C*14**	2	0.98	34	3.7	0.05	0.26	[0.03—1.02]
**C*15**	12	5.88	74	8.1	0.38	0.71	[0.34—1.35]
**C*16**	6	2.94	0	0.0	1	60	[3.37—1069.83]
**C*17**	4	1.96	10	1.1	0.3	1.81	[0.41—6.36]
**C*18**	1	0.49	0	0	1	13.5	[0.55—332.86]
**HLA–DRB1**							
**DRB1*01**	21	10.29	1283	6.7	0.05	1.6	[0.96—2.53]
**DRB1*03**	**21**	**10.29**	**1110**	**5.8**	**0.01**	**1.86**	**[1.12—2.95]**
**DRB1*04**	12	5.88	1838	9.6	0.07	0.59	[0.3—1.05]
**DRB1*07**	13	6.37	1513	7.9	0.51	0.79	[0.41—1.39]
**DRB1*11**	61	29.9	5169	27	0.34	1.15	[0.84—1.57]
**DRB1*13**	15	7.35	1895	9.9	0.29	0.72	[0.4—1.23]
**DRB1*14**	14	6.86	1149	6	0.55	1.15	[0.62—1.99]
**DRB1*15**	13	6.37	1455	7.6	0.6	0.83	[0.43—1.45]
**DRB1*16**	28	13.73	2259	11.8	0.38	1.19	[0.77—1.78]
**HLA–DQB1**							
**DQB1*02**	31	15.35	28	11.2	0.26	1.41	[0.79—2.54]
**DQB1*03**	82	40.59	0	0	1	0	[0—Inf]
**DQB1*05**	**62**	**30.69**	**54**	**22**	**0.04**	**1.57**	**[1.01—2.46]**
**DQB1*06**	25	12.38	30	12.2	1	1.02	[0.55—1.86]
**DQB1*07**	0	0	71	28.8	1	0	[0—Inf]
**DQB1*08**	0	0	9	3.8	1	0	[0—Inf]

Note: statistically significant associations are presented in bold. Abbreviations: cHBV—chronic hepatitis B virus; CI—confidence interval; HLA—human leucocyte antigen.

## Data Availability

The original data presented in the study are openly available in FigShare at [DOI: 10.6084/m9.figshare.28577360].

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
