# Peer review of "Chronic Hepatitis B Virus Infection and HLA Variations in a Greek Population"

_viruses, 2025, doi:10.3390/v17040462_

Round 1
Reviewer 1 Report
Comments and Suggestions for Authors
Great efforts have been made to identify associations between host immunogenetics and chronic HBV infection. A number of host genetic variants, such as mutations in human leukocyte antigens (HLAs), cytokine and chemokine genes, toll-like receptor (TLRs), have been found to influence the outcome of HBV infection [Zhang et al. Host Genetic Determinants of Hepatitis B Virus Infection. Front Genet. 2019 Aug 13;10:696. doi: 10.3389/fgene.2019.00696; Seshasubramanian et al. Human leukocyte antigen A, B and Hepatitis B infection outcome: A meta-analysis. Infect Genet Evol. 2018 Dec;66:392-398. doi: 10.1016/j.meegid.2017.07.027]. The aim of Myserli et al. manuscript was to study HLA alleles' impact on HVB-infected individuals in a Greek population: 107 patients with chronic HBV infection and 101 with spontaneous clearance. The study showed an association of 9 specific HLA alleles with either susceptibility or protection against chronic HBV infection. Limitations of the study included the small sample size.
Remarks
- Table 1 does not indicate the demographic characteristics of the group of healthy subjects.
- Since there is a lot of data on the polymorphism of host genes, it is difficult to understand the novelty of the data obtained by the authors. Therefore, it is necessary to clearly write in the Discussion and/or in the Conclusions what new results the authors have shown in their work.
- It is probably more correct to compare all 3 groups with each other. It is unclear how healthy people differ from patients with spontaneous hepatitis B clearance, but some HLA alleles' frequencies vary significantly.
Reviewer 2 Report
Comments and Suggestions for Authors
This paper provided the HLA allleles' fequencies across chronic HBV infection group, spontaneous HBsAg-negative group and health blood donors. The major concern is that the cohort of patient is much smaller than the healthy cohort, which may lead to errors in the conclusions.
Round 2
Reviewer 2 Report
Comments and Suggestions for Authors
All my concerns have been addressed, thanks.